# Olive Fruit Selection through AI Algorithms and RGB Imaging

**DOI:** 10.3390/foods11213391

**Published:** 2022-10-27

**Authors:** Simone Figorilli, Simona Violino, Lavinia Moscovini, Luciano Ortenzi, Giorgia Salvucci, Simone Vasta, Francesco Tocci, Corrado Costa, Pietro Toscano, Federico Pallottino

**Affiliations:** 1Consiglio per la Ricerca in Agricoltura e l’Analisi dell’Economia Agraria (CREA)—Centro di Ricerca Ingegneria e Trasformazioni Agroalimentari, Via della Pascolare 16, Monterotondo, 00015 Rome, Italy; 2Dipartimento Di Ingegneria Civile e Ingegneria Informatica, Università Degli Studi di Roma “Tor Vergata”, Via del Politecnico 1, 00133 Rome, Italy; 3Consiglio per la Ricerca in Agricoltura e l’Analisi dell’Economia Agraria (CREA)—Centro di Ricerca Ingegneria e Trasformazioni Agroalimentari, Via Milano, 43, 24047 Treviglio, Italy

**Keywords:** olive classification, colour calibration, conveyor belt, CNN model, machine learning

## Abstract

(1) Background: Extra virgin olive oil production is strictly influenced by the quality of fruits. The optical selection allows for obtaining high quality oils starting from batches with different qualitative characteristics. This study aims to test a CNN algorithm in order to assess its potential for olive classification into several quality classes for industrial purposes, specifically its potential integration and sorting performance evaluation. (2) Methods: The acquired samples were all subjected to visual analysis by a trained operator for the distinction of the products in five classes related to the state of external veraison and the presence of visible defects. The olive samples were placed at a regular distance and in a fixed position on a conveyor belt that moved at a constant speed of 1 cm/s. The images of the olives were taken every 15 s with a compact industrial RGB camera mounted on the main frame in aluminum to allow overlapping of the images, and to avoid loss of information. (3) Results: The modelling approaches used, all based on AI techniques, showed excellent results for both RGB datasets. (4) Conclusions: The presented approach regarding the qualitative discrimination of olive fruits shows its potential for both sorting machine performance evaluation and for future implementation on machines used for industrial sorting processes.

## 1. Introduction

Olive oil (*Olea europaea* L.) is a key ingredient of the Mediterranean diet [1]. Extra virgin olive oil (EVOO) is produced in Italy in the same quantity in which it is exported. Oil produced by other countries is not exported from Italy precisely because this country has a worldwide reputation for Made in Italy olive oil [2]. Certainly the increase in consumption of this product over the years is due to its quality, in terms of nutritional properties [3].

To maintain the chemical and sensory quality standards of EVOO, it is important to process the harvested olives quickly and to pay special attention during mechanical harvesting to avoid internal damage, which could compromise the quality of the final product [4].

Excellent olives are necessary to produce high-quality oil. In this regard, to increase batch quality, and the sorting and grading of products, optoelectronic machines are widely implemented. Automatic sorting is known to save time and reduce manual labor. Several studies have been conducted on image-based automatic sorting of agri-food products [5].

In the food industry, two sorting techniques (manual and mechanical) are used to identify and classify products. For example, to identify specific characteristics of coffee beans, the use of mechanical sorters allows selection by size [6]. As reported by Haff and Pearson [7], a sorter using NIR spectroscopy was used to sort and classify pistachios according to defects.

Regarding olives, in the study by Babanatis–Merce et al. [8], an optical sorter with an RGB sensor was used to determine the color of the olives analyzed, but only considered 100 black olives and 100 green olives to assess their ripeness.

The key aspect of the sorting process is the implementation of advanced algorithms applied for selection. Today, artificial intelligence algorithms are the most useful approach to identify and classify food products because they are both fast and reliable. For example, deep neural networks (DNNs) are characterized by many layers of internal processing, combining the ability to recognize objects based on color, texture and shape. There have been many studies on the use of convolutional neural networks (CNNs), including the fast and efficient algorithms available from the You Only Look Once (YOLO) family. These methods unify the classification and localization of targets in a regression problem, without needing proposed networks per region, and directly perform regression to detect targets from the image [9]. Machine learning algorithms find several fields of application in the agricultural panorama. In the study by Ghazanfari et al. [10], artificial neural networks were used to classify four Iranian pistachio cultivars based on their 2D shapes. They used shape recognition methods to classify pistachios into four different classes. Typically, these algorithms are used in the pre-production stage, particularly to predict crop yield, soil properties and irrigation requirements. The use of these technologies in precision agriculture enables farmers to support crop decisions and improve site-specific and smart farming practices. In the work of Ben Ayed and Hanana [11], 18 table olive cultivars from around the world were analyzed considering different morphological, biological and physicochemical parameters, and the Bayesian network was applied to evaluate the influence of these parameters on tolerance, productivity and oil content. As a result, oil content was shown to be strongly influenced by crop tolerance. Despite their usefulness, artificial intelligence algorithms may present object detection problems. These methods are highly dependent on light conditions and on the device used for acquisition. For these reasons, some methods have been developed to standardize the signal [12].

With the advent of CNNs, and YOLO in particular, the problem of signal standardization seems to be solved. In fact, these algorithms are trained on such large data sets that, in principle, they have encoded all possible light conditions. However, CNNs seem to fail in the recognition of dark images [12]. This is rather surprising, because if the network is trained on dark images, in principle, it should be able to recognize them.

Some studies in literature show prior studies concerning the use artificial intelligence algorithms for olive classification. Aquino et al. [13] developed an artificial-vision algorithm able to classify images taken in the field for the identification of olives directly from the tree in order to collect data for accurate predictions on yield. Other studies [14] use RGB image acquisition and CNNs for the early estimation of ripening stages of fruits still on-branch closely related to the quality and quantity of orchard production. In the work by Furferi et al. [15], an Artificial Neural Network (ANN)-based method was proposed for automatic Ripening Index (RI) evaluation. The process was adopted considering four different classes (“green,” “spotted,” “purple” and “black” stages) assigned by olive skin and pulp color evaluation, without considering the presence of defects. Instead, in [16] the olive classification recognized fruits harvested from the tree or the ground using a static computer vision system. In this study [16], the image classification was conducted using a shallow learning method by means of an ANN with a single hidden layer.

As reported by Benos et al. [17], enormous progress has been made in machine learning in recent years. In particular, during the years 2018–2020, there was a 745% increase in the number of articles on machine learning related to agriculture, as a variety of machine learning algorithms are used on crops and animals, based on input parameters from satellites and drones.

Generally speaking, machine learning (ML) models are well-used within the food sector, e.g., for safety monitoring and prediction purposes. Currently, there are some works on the application of ML in food safety, particularly on foodborne pathogens and diseases or applications of ML to trace the source of foodborne diseases [18].

Another example food safety concerns beer quality. In the study by Helfer et al. [19], NIR spectral data from 60 beer samples were obtained and analyzed using Raspberry Pi 3 devices connected to a network running machine learning. The results showed that by applying machine learning and Raspberry Pi 3 devices in parallel, we are able to improve performance (reduce the time by 57% in order to find the best regression coefficient) to produce more efficient and predictive models.

Considering the present bibliography, this study aims to test a CNN algorithm in order to assess its potential for olive classification into several quality classes for industrial purposes, including its potential integration and sorting performance evaluation. Images were acquired by a conveyor belt equipped with an RGB camera. An advanced colorimetric calibration algorithm was applied to fortify the results. The best performing algorithm has been applied on olive lots selected by an industrial sorting machine at different speeds and classes, to evaluate the potentiality and efficiency of the algorithm. Unlike previous studies, this study attempts the classification using a CNN algorithm, a conveyor belt and expert evaluation for comparison.

## 2. Materials and Methods

### 2.1. Olive Data

Narducci Oil Mill, located in Moricone (central Italy), a member of the SABINA DOP Consortium, supplied freshly harvested olives to carry out the olive selection. The two olive cultivars considered were Salviana and Leccino, which are typical of the region of Sabina (Lazio, Italy). The Leccino cultivar produces an oil with a fresh flavor. Leccino is the most widely grown cultivar in the world, as it combines excellent productivity and oil yield with exceptional grace, as well as adequate cold resistance [20].

The Salviana cultivar is grown exclusively in three municipalities of lower Sabina. Although extremely rare today, it was once widely grown, so much so that it has characterized the history and culture of the Sabine territory since the time of the ancient Romans [21]. On 2 November (season 2021), the olives were picked with twigs and wet leaves, for a gross weight of 929 kg. Then the fruits were cleaned, and 845 kg were collected to perform the tests (on 4 November).

### 2.2. Expert Olive Evaluation

To evaluate the performance of an industrial RGB optical sorting machine, prior manual training is needed. Before the automated selection, a trained expert operator classifies the samples, dividing them into five different classes based on color and presence of damage. Defects, in terms of integrity, coloring and consistency, are attributable to mechanical factors, which cause percussion or crushing injury, or biotics, caused by insects (e.g., *Bactrocera*), bacteria (e.g., *Pseudomonas*) or fungi (e.g., *Gloeosporium*). Figure 1 shows some samples for each class (A–F). The five classes considered are named: “Top Green”(A), “Good Green”(B), “Good Black”(C), “Bad Green”(D–E) and “Bad Black”(F). The “Top Green” class identifies completely green olives without defects; the “Good” category identifies intact and oleic olives and the “Good Green” class includes also lightly invariate fruits; the “Bad” category identifies damaged olives, not good for EVOO production. Currently, bad olives are commonly used for oil production, but it may alter the oil’s characteristics and quality [22].

### 2.3. Conveyor Belt Prototype

To develop algorithms for the selection of olives, a pilot conveyor belt based on image analysis was used (Figure 2).

The low-cost prototype used open-source technologies, and it relayed image analysis for the extraction of three main qualitative attributes: size, shape and color. The system and algorithms are also useful for the analysis of many other agri-food products [23]. The prototype was equipped with an imaging sensor for continuous scanning and was composed of a variable speed conveyor belt (W∗L∗H 40∗205∗5 cm^3^) driven by an asynchronous AC motor, a feeding hopper and a rotary encoder (made by Eltra, model Eltra RH200B) with 1024 ticks per revolution, operating at 5 volts/20 mA per channel and connected with 4 control pins. The signal from the encoder was acquired through a custom-made circuit based on the open-source electronic prototyping platform Arduino™ [24]. The signal mentioned above was also processed to achieve quadrature of the encoder to define the direction of rotation. This method leads to fewer errors in the reading and an increase of ticks to 4096 per revolution. The conveyor belt moved at a constant speed of 1 cm/s, allowing olive sample image acquisition every 15 s. Images were acquired by a Mako RGB camera (model Mako G, Allied Vision, Stadtroda, Germany) placed on an aluminum frame provided with a Complementary Metal-Oxide Semiconductor (CMOS) sensor with a resolution of 1456 pixels (width) × 1088 pixels (height), with a framerate of 30 Hz (30 frames per second). To avoid alteration from external light sources, the camera was covered by a box equipped with controlled Light Emitting Diode (LED) strips to uniformly illuminate the samples. Therefore, the camera was connected to a PC, which ran software made in Java based on multithreading programming structured to manage image acquisition settings, general settings, graphical user interface and serial acquisition of multiple files [25].

The images were also calibrated following the Thin-Plate Spline interpolation function [26] in the red, green, and blue (RGB) space. As reported by Menesatti et al. [27] the ColorChecker X-Rite 24 patches, reporting known sRGB coordinates, was included in each acquired image. Afterwards, the measured ColorChecker sRGB coordinates within each image (i.e., considering its whole field) were warped (transformed) into the reference coordinates of the same ColorChecker. The procedure was elaborated in Matlab 7.1 R14 modifying the 2-d TPS code by Ossadtchi [28].

### 2.4. Artificial Intelligence (AI) and Conventional Neural Network (CNN)

A total of 557 olives from the five aforementioned classes were analyzed. Images were manually labelled. Data set composition and population are reported in Table 1.

Before training, the data sets of calibrated and non-calibrated images were randomly split into training and test set with a ratio of 70 to 30%.

Figure 3 shows a paradigmatic comparison between calibrated (A–B) and non-calibrated (C–D) images.

Transfer learning using Alexnet as the pre-trained network, as included in the deep learning MATLAB (R2019b) toolbox, was implemented. The net was trained with the options reported in Table 2 using stochastic gradient descent with momentum as the optimizer and reshuffling the dataset every epoch.

This was utilized because the mini-batch size does not evenly divide the number of training samples. No data augmentation was applied during the training process. Since the final output was obtained by means of a softmax function and is a 5-dimensional probability vector p, whose components represent the output probability of the corresponding class, the output class is given by the lager component of the vector p.

### 2.5. Algorithm Implementation and Industrial Sorting

Once developed and tested on the pilot conveyor belt, the algorithm demonstrated adequate results, as demonstrated in Chapter 3. After observing its performance while acquiring moving objects, we think its utility might benefit industrial sorting machines after proper code streamlining and implementation, and it may also be used to verify the classification performance of specific industrial machine settings.

Figure 4 reports the flowchart regarding olive sorting, with the implementation of the AI algorithm tested in the conveyor belt and its potential implementation in an industrial sorting machine. In fact, sorting machines often use simple colorimetric thresholds (like the one presented), due to the need of very time efficient algorithms.

The same lot of olives used on the conveyor belt to build the CNN models was partially selected by an industrial sorting machine. The pneumatic optoelectronic sorting machine used is the INFINITY Plus model (Italian Sorting Technology, I.S.T. Srl, Ferrara, Italy). The classification of olives by the grader was performed by considering olive maturity (green olives and black olives). The selection speed of the vibrators was set to two different velocities: slow (30%) and fast (55%). A total of 1156 kg of olives were selected for the fast speed and 1800 kg for the slow speed (Table 3).

After displaying the acquired images on the machine’s screen, the operator can set the parameters that best select the desired classes by adjusting the thresholds of RGB values. The images captured by the class can be seen on the debug screen to verify the correct operation of the cameras. The parameters of the solenoid valves, which regulate the ejection of olives, are set as follows: the time the solenoid valve remains open, set to 40 in units of 100 microseconds; the delay between defect detection and valve opening, set to 10 in units of 100 microseconds; and the burst, the number of pixels (set to 3 pixels) that are added to the defect to define how many solenoid valves should be opened. Each image can be edited according to “Green,” “Red” and “Blue” values, choosing the values to increase or decrease for each class, to isolate the color of the defect or to discount it. Once the parameters are set, the classes are created, and the process is repeated on each camera.

The images of a random selection (30 kg for each lot) of the olives selected by the sorting machine were acquired with the conveyor belt. The CNN model was applied to these images to obtain the olives’ classification into the 5 classes: Bad Green, Good Green, Top Green, Bad Black and Good Black. The result of this classification is reported and discussed below.

## 3. Results

Table 4 reports the characteristics and main results of the CNN model used to predict calibrated and non-calibrated images.

The CNN of calibrated and non-calibrated images trained had a total of 8 convolutional layers, and the algorithm converged after 390 iterations.

Training “transfernet” with 10 epochs gave a validation accuracy of 90.4% for calibrated olive images, while a validation accuracy for non-calibrated olive images was 88.0%. Also, r Training shows a greater validation accuracy in calibrated olive images (98.7%) as opposed to 97.9% in non-calibrated olive images.

Figure 5 shows training, test and overall confusion matrices for the calibrated data set. From the Training Confusion Matrix of the calibrated olives, it can be observed that “Bad Black” olives are classified as 98.8% (only one “Bad Black” is classified as “Good Black”).

“Bad Green” olives are classified at 96.4% (only two “Bad Greens” are classified as “Top Green”). “Good Green” olives have a 97.4% classification because two “Good Greens” are classified as “Top Green”. Finally, “Good Black” olives and “Top Green” olives present a 100% classification rate. Only in the Test Confusion Matrix and Total Confusion Matrix are “Bad Black” olives classified as Green (both Bad Green and Good Green), but they are still correctly classified with a percentage of 85.3% and 94.7% for the two tests, respectively. For this reason, “Bad Black” olives classified as Green can be considered a negligible figure, given the model’s high classification rate.

Figure 6 reports training, test and overall confusion matrices for the non-calibrated data set.

Regarding the Training Confusion Matrix of the non-calibrated olive images, the classification is similar to that of the calibrated images. In contrast, both the Test Confusion Matrix and Total Confusion Matrix show a much lower percentage classification than the calibrated images. In particular, “Bad Green” olives are classified as “Good Green” as well as “Top Green” (88.8%). “Good Black” olives are also classified as “Good Green” (86.6%) in the Total Confusion Matrix. Figure 7 shows an example of the olives’ classification through the transfernet learning.

The results of the algorithm, tested to evidence the selection performance of the cited sorting machine using two different selection speeds, are reported in Table 5 and Table 6.

Table 5 reports the results regarding the calibrated images, while Table 6 reports those of the non-calibrated ones. When compared, it is possible to see that the algorithm, if used on calibrated images, leads to different percentages of classification. For example, a substantial increase of the percentage of classification of the good black and good green olives is noticeable.

## 4. Discussion

The performance between fast and slow in the images of calibrated olives in selecting green to black is similar; 65% of black olives (Bad or Good) were classified by the sorting machine as black, regardless of speed; 88% of green olives (Bad, Good or Top) were classified by the sorting machine as green, regardless of speed.

With regard to non-calibrated olive images, the performances between fast and slow are similar; 65% of the black olives (bad or good) were classified by the sorting machine as black at the slow speed; while at the fast speed, 63% of the black olives (bad or good) were classified by the sorting machine as black. A total of 88% of the green olives (bad, good or top) were classified by the sorting machine as green at the slow speed and 89% at the fast speed.

In the images of the non-calibrated olives, “Bad Black” and “Good Black” have higher percentages than in the calibrated images, at 19.7% and 23.4% respectively.

It is still possible to discard bad black olives (10%) and bad green olives (17%), which add defects to the oil [22].

By applying one of the two CNN models on calibrated or non-calibrated images, it is possible to obtain oils of different quality [29], for example:Superior EVOO (“Top green” olives only) using 17% of total olives;High quality oil obtained by green olives (only “Top green” and “Good green” olives) using 40% of the total olives;High quality oil (produced by only “Top green,” “good green” and “good black” olives) using 72% of the total olives.

In literature, studies that analyze the olive oil produced from each of the selected olive lots were not observed. However, there are studies that report experiences related to classification algorithms applied to olive fruit classification.

Some studies examined how to estimate the maturity stage of olives [30] or detect bruised fruits [31] through image analysis. However, unlike our study, they adopted static conditions for acquisition and used segmentation for an easy detection of edges from the background. Additionally, Ponce et al. [32] focused on olive fruits analysis throughout segmentation to automatically count and evaluate size and fruit mass. Some other works, e.g., Aguilera Puerto et al. [33] used conveyor belt systems paired to AI. In this case, the application regarded the discrimination between olives picked from the ground from those picked from the tree, before and after cleaning, using a Multilayer Perceptron (MLP) neural network for class detection. This method demonstrated good results (success rate of 98.8%, after washing), but there were no applications for industrial machines for the physical sorting of the fruits and evaluation of the oil obtained.

Regarding the employment of CNN in the olive sector, Ponce et al. [34] classified olives on the basis of their variety: the image acquisition was conducted in a static box with artificial lighting to isolate objects from background. Different CNNs (AxelNet, Inception and ResNet) were trained and tested for distinguishing seven varieties. However, the training dataset was composed of binary images and rotation-based data augmentation, discarding features related to color variability. Therefore, the method used in this study introduces a novelty, developing an AI algorithm for detection of both maturity stage and defects through a CNN-based approach, testing the output of an industrial sorting machine.

## 5. Conclusions

To concluding, the results show that the colorimetric calibrated images have a slightly better performance of selection compared to the non-calibrated ones, therefore this method could be introduced and adopted to boost physical olive fruit sorting processes. The trained and applied algorithm represents a potential valid tool to implement in conventional sorting machines. Indeed, many sorting machines use conventional image analysis algorithms and could benefit from the introduction of AI models. However, the algorithm needs additional work regarding the streamlining and implementation of the code. A challenging aspect of industrial processing includes the elaboration time, which is very limited. Future studies will attempt to test the algorithm within an industrial machine to physically compare the selection performances with those obtained through simple image analysis elaboration techniques. Finally, as shown by the oils produced and evaluated by trained panelists, these machines produce substantially different oils. However, future studies could attempt to vary the selection parameters, with the aim to produce oils with different characteristics of the sensorial profile, which is required more and more frequently by the final consumer.

## Figures and Tables

**Figure 1 foods-11-03391-f001:**
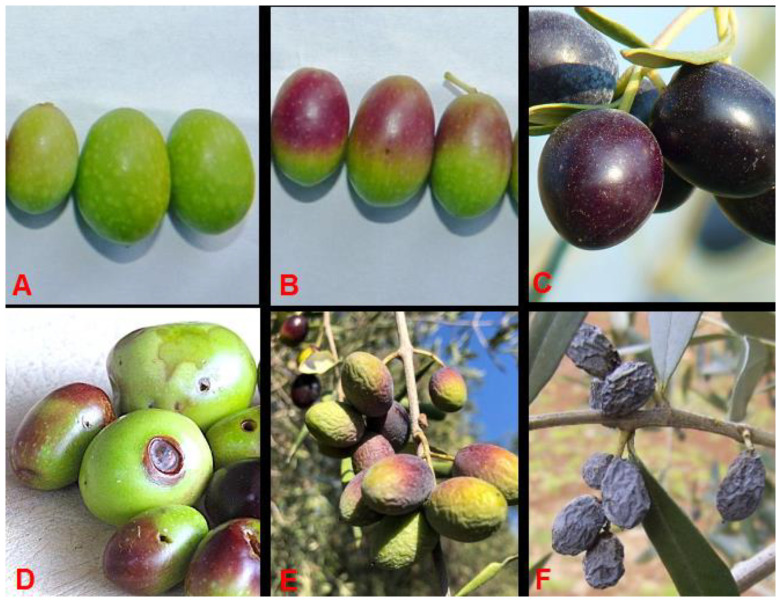
(**A**) “Top Green;” (**B**) “Good Green;” (**C**) “Good Black;” (**D**,**E**) “Bad Green;” (**F**) “Bad Black”.

**Figure 2 foods-11-03391-f002:**
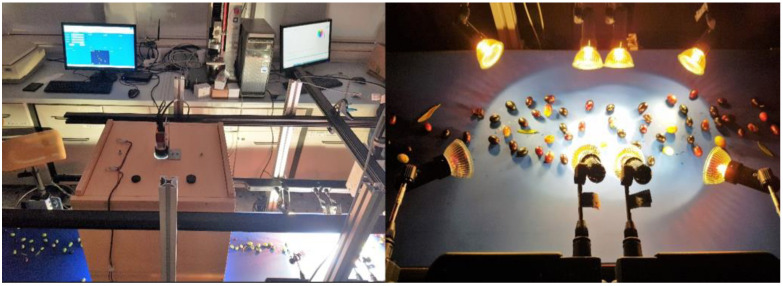
A pilot conveyor belt used to select olives.

**Figure 3 foods-11-03391-f003:**
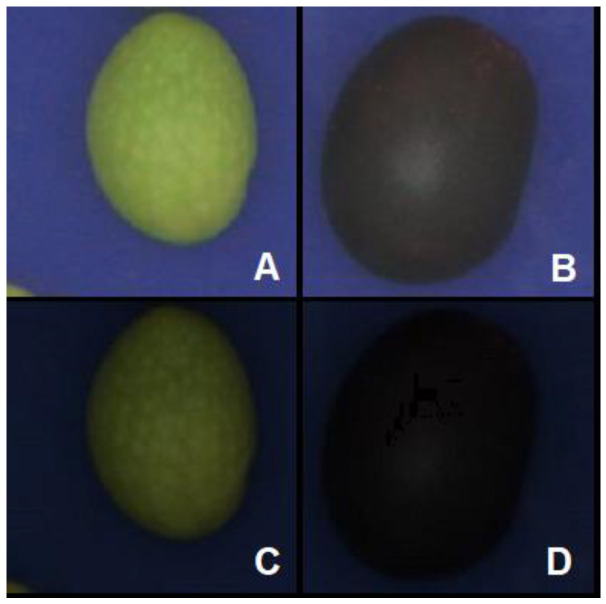
(**A**,**B**) Comparison between calibrated and (**C**,**D**) non-calibrated images of olives belonging to the class “Top Green” and “Good Black”.

**Figure 4 foods-11-03391-f004:**
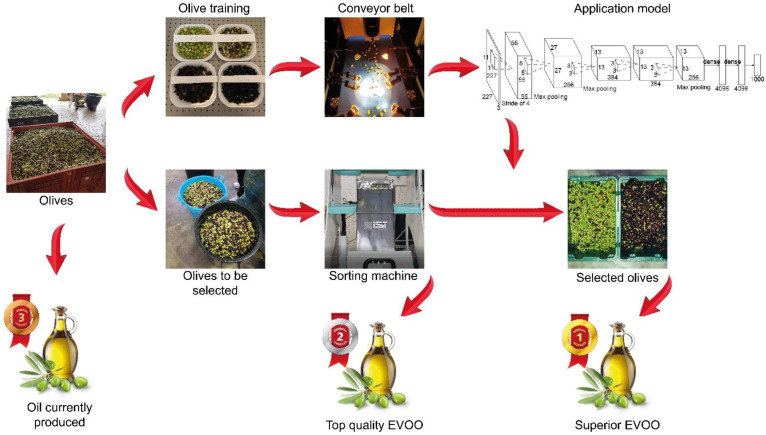
Implementation of the AI algorithm in the conveyor belt and a potential implementation in an industrial sorting machine.

**Figure 5 foods-11-03391-f005:**
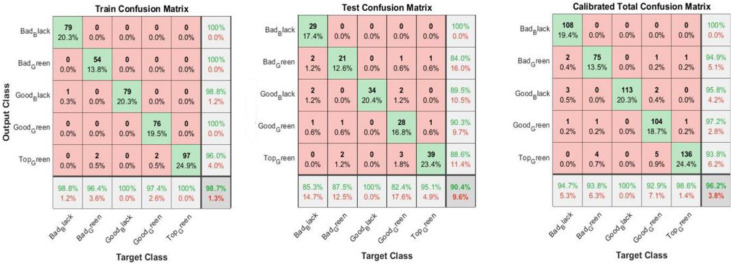
Confusion matrices for training, test and overall calibrated data sets. The off-diagonal elements of these matrices show the number of false positive (**up right**) and false negative (**down left**) cases.

**Figure 6 foods-11-03391-f006:**
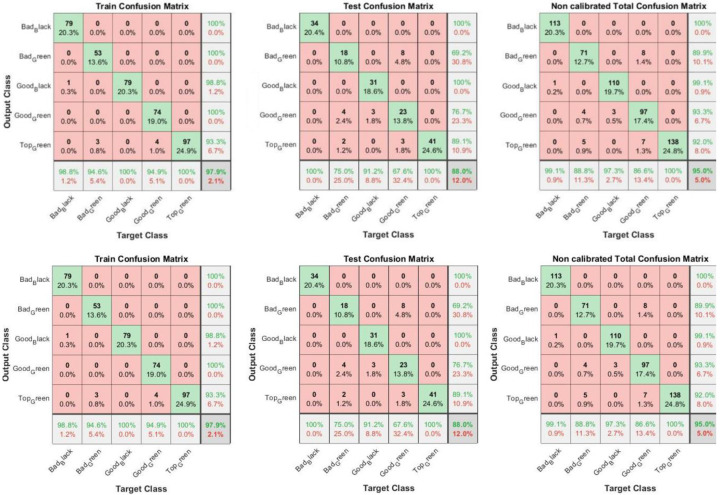
Confusion matrices for training, test and overall non-calibrated data sets. The off-diagonal elements of these matrices show the number of false positive (**up right**) and false negative (**down left**) cases.

**Figure 7 foods-11-03391-f007:**
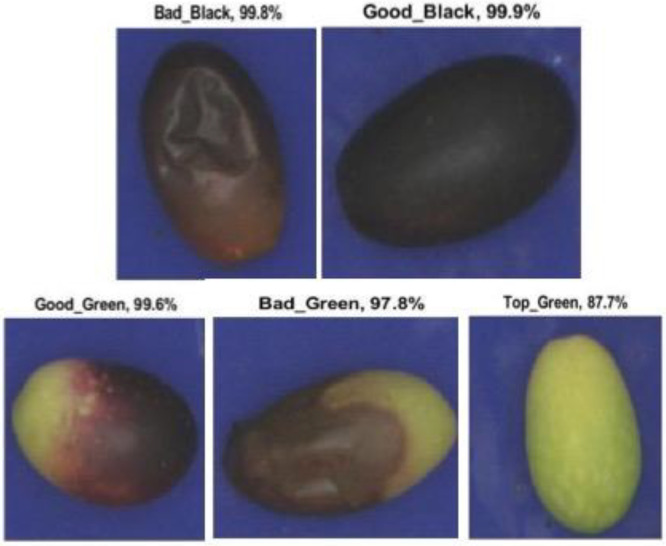
Example of olives’ classification through the transfernet learning.

**Table 1 foods-11-03391-t001:** Data set olives composition.

Olive Classes	Bad Black	Good Black	Bad Green	Good Green	Top Green
N. Samples	114	113	80	112	138

**Table 2 foods-11-03391-t002:** Net training settings.

Optimizer	(SGDM)
Mini batch size	10
Maximum epochs	10
Initial learning	10^−4^
Validation frequency	6

**Table 3 foods-11-03391-t003:** Weight of olives based on speeds (slow and fast).

Olive Classes	Weight (Slow Speed)	Weight (Fast Speed)
Green	946 kg	433 kg
Black	854 kg	723 kg
Total	1800 kg	1156 kg

**Table 4 foods-11-03391-t004:** Results of the CNN model for calibrated and non-calibrated images.

CNN Descriptors	Calibrated Values	Non-Calibrated Values
Number of samples	557	557
Number of convolutional layers	8	8
Training time	26″40′	23″35′
% Training set	70	70
r Training	98.7	97.9
r Test	90.4	88.0

**Table 5 foods-11-03391-t005:** Results of the external test of calibrated olives images. The rows report the sorting machine output, while the columns report the conveyor belt results.

Calibrated Olive Images	Bad Black	Bad Green	Good Black	Good Green	Top Green	Total
Black slow	187	81	531	281	21	1101
Green slow	28	266	63	132	283	772
Black fast	60	36	262	127	10	495
Green fast	10	89	25	67	153	344
Percentages	10.5	17.4	32.5	22.4	17.2	100.0

**Table 6 foods-11-03391-t006:** Results of the external test of non-calibrated olive images. The rows report the sorting machine output, while the columns report the conveyor belt results.

Non-Calibrated Olive Images	Bad Black	Bad Green	Good Black	Good Green	Top Green	Total
Black slow	392	80	333	269	27	1101
Green slow	44	221	48	130	329	772
Black fast	88	54	228	115	9	495
Green fast	12	83	25	51	173	344
Percentages	19.7	16.1	23.4	20.9	19.9	100.0

## Data Availability

Data is contained within the article.

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
