# Peer review of "Olive Fruit Selection through AI Algorithms and RGB Imaging"

_foods, 2022, doi:10.3390/foods11213391_

Round 1
Reviewer 1 Report
The main strong points of the work are: (1) the theme is current and strategic involving neural networks, imaging treatment and olive selection; (2) the evaluation used a practical prototype and results presented are relevant. The main weak points are: (1) the discussion of related works was more descriptive than comparative (part of section 1); (2) as the text does not present a comparison of the proposal with other approaches, it is difficult to evaluate the relevance of the scientific contribution.
I found aspects to be considered in a future version of the article. I will list them in the same order they appeared during the reading. I hope the comments will be used to improve the research and the text. At the end, I will present my final evaluation.
1) The abstract must be improved to clarify the scientific contribution of the article. The authors discuss several aspects of the work, but what is the difference in relation to related works (state-of-the-art)? The reader ends the abstract without know what is the scientific contribution of this research.
2) It is important to include in the section "Introduction", a more complete description of the paper’s proposal. The section discusses several general topics, but the most important aspect of a scientific paper is not presented, namely, the scientific contribution (its difference in relation to state of the art). The authors included the sentence “No work is reported in the literature concerning the industrial selection of olives for 83 EVOO production and the use of artificial intelligence algorithms for olive classification”. This sentence indicates that the main contribution seems to be the application in olive selection and no regarding the technological solution involving machine learning. I recommend improving the discussion of works to explain the scientific contribution of this proposal. I also recommend the inclusion of the “Research Question” that guided the study;
3) In addition, I recommend including in the “Introduction” general references related to the use of machine learning on analysis regarding food and drink. In this sense, the authors can use current references to renew the discussion of food and machine learning, not specifically focusing on olive selection. I recommend citing three recent references related with this generic theme: (1) Machine Learning in Agriculture: A Comprehensive Updated Review, Sensors, 2021. https://doi.org/10.3390/s21113758; (2) The application of parallel processing in the selection of spectral variables in beer quality control. Food Chemistry, 2022. https://doi.org/10.1016/j.foodchem.2021.130681 (3) Application of machine learning to the monitoring and prediction of food safety: A review. Comprehensive Reviews, 2021. https://doi.org/10.1111/1541-4337.12868
4) I believe the following aspect is the most relevant weakness of the paper. There is a discussion of related works in Section 1, but I believe that it would be interesting to improve the organization through a “comparison with the proposed work”. Typically, based on a revision of related works is possible to indicate the contribution sought by an article. Section 1 discusses some related works, but the section does not indicate how the works were found (Did you use a “search string”?) and it also does not have a comparison with this research. It is important to choose comparison criterions and use them to compare and discuss the related works. I advise the authors to include a comparison table to discuss the characteristics of the works. Based on this table, it would be easier to understand the relation between them, their relationship with the proposal and the scientific contribution of the work. I recommend to organize the discussion of related works in a specific section dedicated to this, maybe a section 2 called “Related Works”;
5) Section 2 addresses “Materials and Methods”. The section describes the strategy used to organize the study. The text indicates relevant practical aspects that supported the study such as the use of data from real freshly olives (subsection 2.1) and how experts participated in the preparation of the material (subsection 2.2). In addition, subsection 2.3 presents a prototype used in the work. The prototype is a relevant aspect of the wok because it enriches the proposal with a concrete vision. But, I recommend to better describe the prototype with a figure showing its organization (parts and relation between them). The text has only a figure with photos that do not explain its organization. It is not easy to understand the system only with photos. Subsection 2.5 indicates a “possible implementation” in industry. I did not understand the relevance to present this vision. I recommend the authors to better explain the relevance of that subsection. Was the prototype really integrated into an industrial environment?
6) Section 3 presents the approaches “Results”. The section is well organized and the results are relevant. However, I recommend to better discuss the tables 5 and 6. I believe the discussion of those tables can be improved to help the readers to understand the results;
7) Section 4 contains a discussion of the results. The section is small and I believe the authors can improve the text through a discussion of lessons learned in the research. I recommend to create a table with the lessons learned and discuss them in the text;
8) I advise the authors to improve the "Conclusion" (Section 5) through a better discussion of main findings using concrete results produced in the study. Finally, the authors can improve the text with a better discussion of future works, I found only one simple sentence regarding this.
Based on these comments, I can present my final evaluation. I think the article will need a careful revision and after that, the article can be reviewed again. I consider that paper will need “Major Revision”.
Author Response
REVIEWER 1
The main strong points of the work are: (1) the theme is current and strategic involving neural networks, imaging treatment and olive selection; (2) the evaluation used a practical prototype and results presented are relevant. The main weak points are: (1) the discussion of related works was more descriptive than comparative (part of section 1); (2) as the text does not present a comparison of the proposal with other approaches, it is difficult to evaluate the relevance of the scientific contribution.
Thank you for outlining the strengths of the work. Bibliography was deepened and introduced in the introduction and discussion of the paper.
I found aspects to be considered in a future version of the article. I will list them in the same order they appeared during the reading. I hope the comments will be used to improve the research and the text. At the end, I will present my final evaluation.
Thank you.
1) The abstract must be improved to clarify the scientific contribution of the article. The authors discuss several aspects of the work, but what is the difference in relation to related works (state-of-the-art)? The reader ends the abstract without know what is the scientific contribution of this research.
Thanks for pointing that out, some related works have been introduced and the abstract modified.
2) It is important to include in the section "Introduction", a more complete description of the paper’s proposal. The section discusses several general topics, but the most important aspect of a scientific paper is not presented, namely, the scientific contribution (its difference in relation to state of the art). The authors included the sentence “No work is reported in the literature concerning the industrial selection of olives for EVOO production and the use of artificial intelligence algorithms for olive classification”. This sentence indicates that the main contribution seems to be the application in olive selection and no regarding the technological solution involving machine learning. I recommend improving the discussion of works to explain the scientific contribution of this proposal. I also recommend the inclusion of the “Research Question” that guided the study;
The aim has been rewritten to underly its content and additional info are now present.
3) In addition, I recommend including in the “Introduction” general references related to the use of machine learning on analysis regarding food and drink. In this sense, the authors can use current references to renew the discussion of food and machine learning, not specifically focusing on olive selection. I recommend citing three recent references related with this generic theme: (1) Machine Learning in Agriculture: A Comprehensive Updated Review, Sensors, 2021. https://doi.org/10.3390/s21113758; (2) The application of parallel processing in the selection of spectral variables in beer quality control. Food Chemistry, 2022. https://doi.org/10.1016/j.foodchem.2021.130681 (3) Application of machine learning to the monitoring and prediction of food safety: A review. Comprehensive Reviews, 2021. https://doi.org/10.1111/1541-4337.12868
This was done, thank you.
4) I believe the following aspect is the most relevant weakness of the paper. There is a discussion of related works in Section 1, but I believe that it would be interesting to improve the organization through a “comparison with the proposed work”. Typically, based on a revision of related works is possible to indicate the contribution sought by an article. Section 1 discusses some related works, but the section does not indicate how the works were found (Did you use a “search string”?) and it also does not have a comparison with this research. It is important to choose comparison criterions and use them to compare and discuss the related works. I advise the authors to include a comparison table to discuss the characteristics of the works. Based on this table, it would be easier to understand the relation between them, their relationship with the proposal and the scientific contribution of the work. I recommend to organize the discussion of related works in a specific section dedicated to this, maybe a section 2 called “Related Works”;
Thank you for the comment. Since few works are present on the subject, we searched again with a series of different strings and the works found are now discussed in the dedicated section discussion, as planned in the guide for authors.
5) Section 2 addresses “Materials and Methods”. The section describes the strategy used to organize the study. The text indicates relevant practical aspects that supported the study such as the use of data from real freshly olives (subsection 2.1) and how experts participated in the preparation of the material (subsection 2.2). In addition, subsection 2.3 presents a prototype used in the work. The prototype is a relevant aspect of the wok because it enriches the proposal with a concrete vision. But, I recommend to better describe the prototype with a figure showing its organization (parts and relation between them). The text has only a figure with photos that do not explain its organization. It is not easy to understand the system only with photos. Subsection 2.5 indicates a “possible implementation” in industry. I did not understand the relevance to present this vision. I recommend the authors to better explain the relevance of that subsection. Was the prototype really integrated into an industrial environment?
Thank you. The text has been enriched to let the reader better understand the use and scop of the conveyor belt used in the present study. However, the text already report as the conveyor belt is a pilot one, therefore not implementable on an industrial environment. Possible implementation is referred to the algorithm applied for the selection into an industrial sorting machine. The usefulness of the 2.5 section is now clarified.
6) Section 3 presents the approaches “Results”. The section is well organized and the results are relevant. However, I recommend to better discuss the tables 5 and 6. I believe the discussion of those tables can be improved to help the readers to understand the results;
The paragraph has been implemented as requested.
7) Section 4 contains a discussion of the results. The section is small and I believe the authors can improve the text through a discussion of lessons learned in the research. I recommend to create a table with the lessons learned and discuss them in the text;
The section has been improved as requested
8) I advise the authors to improve the "Conclusion" (Section 5) through a better discussion of main findings using concrete results produced in the study. Finally, the authors can improve the text with a better discussion of future works, I found only one simple sentence regarding this.
Thank you, the conclusions have been improved.
Based on these comments, I can present my final evaluation. I think the article will need a careful revision and after that, the article can be reviewed again. I consider that paper will need “Major Revision”.
Thank you.
Reviewer 2 Report
I suggest improving the title to be more understandable.
The Introduction should be more related to the topic of research. More detailed information should be added, and knowledge gaps should be shown. Then, the novelty of the present study should be indicated. Furthermore, not all relevant references are correctly cited in the Introduction.
lines 72-77: Please add the reference.
line 95: How were these two cultivars different? Please provide their characteristics.
line 97: Are these masses for both cultivars?
lines 135-137: Please provide a brief description of the calibration methodology.
line 139: Was the cultivar considered?
Sections 2. Materials and Methods and 3. Results are not fully understandable.
Did the authors use CNN or ANN?
Figures 5 and 6 should be corrected.
line 226: Why do the authors use the term "transfernet" olives?
The discussion should be expanded and include more references.
The conclusions should be more detailed and supported by the results.
Author Response
REVIEWER 2
I suggest improving the title to be more understandable.
Title has been changed we hope it is now clearer.
The Introduction should be more related to the topic of research. More detailed information should be added, and knowledge gaps should be shown. Then, the novelty of the present study should be indicated. Furthermore, not all relevant references are correctly cited in the Introduction.
Thank you for your comment. The bibliography has been deepened in the introduction as requested.
lines 72-77: Please add the reference.
The reference was added [12].
line 95: How were these two cultivars different? Please provide their characteristics.
We added some information about both cultivars.
line 97: Are these masses for both cultivars?
Yes, these are masses for both cultivars this is now cited in the text.
lines 135-137: Please provide a brief description of the calibration methodology.
A brief description has been inserted.
line 139: Was the cultivar considered?
As reported in Materials and Methods section, we considered Salviana and Leccino cultivars.
Sections 2. Materials and Methods and 3. Results are not fully understandable.
The sections have been modified.
Did the authors use CNN or ANN?
Authors used CNN, the text has been cleared.
Figures 5 and 6 should be corrected.
As suggested, we corrected Figure 5 and 6. if you have other change requests identify what kind of changes.
line 226: Why do the authors use the term "transfernet" olives?
The sentence and Figure 7 caption have been modified.
The discussion should be expanded and include more references.
This was done.
The conclusions should be more detailed and supported by the results.
This was done.
Round 2
Reviewer 1 Report
I revised the new version of the article and the response letter.
I consider that the authors conducted an adequate revision.
I have only one recommendation to the final version. The authors indicated "This was done, thank you." in my third observation regarding new references, but they only cited one of the three that I have indicated. I recommend to include the other two because they are related with the theme (food and images) of the paper and they were published in strategic journals.
I am satisfied with the revision and I only recommend this final imprrovement regarding references.
Author Response
Thank you for your suggestions. Now, we added the two references as you suggested.
Reviewer 2 Report
The manuscript has been improved
Author Response
Thank you.